# Acetylated Proteomics of UV-B Stress-Responsive in Photosystem II of *Rhododendron chrysanthum*

**DOI:** 10.3390/cells12030478

**Published:** 2023-02-01

**Authors:** Meiqi Liu, Qi Sun, Kun Cao, Hongwei Xu, Xiaofu Zhou

**Affiliations:** Jilin Provincial Key Laboratory of Plant Resource Science and Green Production, Jilin Normal University, Siping 136000, China

**Keywords:** *Rhododendron chrysanthum* Pall., acetylated proteomics, UV-B stress, photosystem II

## Abstract

*Rhododendron chrysanthum* (*Rhododendron chrysanthum* Pall.), an alpine plant, has developed UV-B resistance mechanisms and has grown to be an important plant resource with the responsive capacity of UV-B stress. Our study uses acetylated proteomics and proteome analysis, together with physiological measurement, to show the *Rhododendron chrysanthum* seedling’s reaction to UV-B stress. Following a 2-day, 8-h radiation therapy, 807 significantly altered proteins and 685 significantly altered acetylated proteins were discovered. Significantly altered proteins and acetylated proteins, according to COG analysis, were mostly engaged in post-translational modification, protein turnover, and chaperone under UV-B stress. It indicates that protein acetylation modification plays an important role in plant resistance to UV-B. The experimental results show that photosynthesis was inhibited under UV-B stress, but some photosynthetic proteins will undergo acetylation modification, which can alleviate the UV-B damage of plants to a certain extent. These results will serve as the basis for more research into the intricate molecular mechanisms underlying plant UV-B adaptation.

## 1. Introduction

Although it only makes up a minor portion of all UV light, UV-B (280–315 nm) has traditionally been viewed as a stress since it could cause a variety of harmful consequences in plants, such as a decrease in the growth rate, partial restriction of photosynthesis, alterations in plant physiology, and peroxidation [1,2,3,4]. Due to their sessile nature, plants are inevitably exposed to UV-B radiation, and they adapt to changes in their environment. Adaptive traits in plants can be seen in changes in plant structure and physiological characteristics [5,6], at the biochemical and genetic levels [7,8]. Chloroplast, the organelle that controls photosynthesis, is indeed highly responsive to UV-B exposure [9]. Several studies have shown that UV-B stress, which disrupts the natural catabolic capacity of plants, is mainly caused by PSII located on the thylakoid membrane of the chloroplasts [10,11].

Photosystem II (PSII) is a photosynthetic unit in the thylakoid membrane, which contains two light-harvesting complexes and a light reaction center. PSII is one of the cellular components that has been shown to be most vulnerable to external stressors [12,13]. UV-B radiation has been reported to severely weaken PSII and destroy its chemical compound [14]. The role of numerous thylakoid membrane proteins, such as D1, D2, CP43, LHCII, and PsbH in the photosynthetic machinery, has been well known. Research has demonstrated that UV-B stress not only decreased chlorophyll fluorescence photosynthetic parameters [15] but also dramatically reduced Fv/Fo as well as Fv/Fm [16].

In previous reports, due to UV-B radiation, the Fv/Fm of oat (*Avena sativa* L.) was inhibited, suggesting the photoinhibition alterations in the PSII of the oat [17]. The ratio of Fv/Fm in two grapevine (*Vitis vinifera* L.) cultivars decreased significantly under UV-B stress [15] in a study of winter wheat (*Triticum aestivum* L.) over ten days. At the same time, after intense UVB treatment, low temperatures aggravated the photoinhibition induced by UVB, and the results showed that both photosynthetic rate (P_N_) and Fv/Fm decreased [18]. The study found that the Fv/Fm of cotton (*Gossypium hirsutom* L.) decreased under UV-B stress, and net photosynthesis was also reduced. In contrast, fluorescence measurements showed that both the Fv/Fm ratio and the reduction capacity were improved without UV-B stress [19].

In our earlier research, we compared the impact of UV irradiation on *Rhododendron chrysanthum* photosynthesis. The findings indicated that the photosynthetic activity of *Rhododendron chrysanthum* was inhibited by UV-B but unaffected by UV-A, and that the inhibitory effects of UV-B were dose dependent [20].

To understand the complex mechanisms of chloroplast formation and the plant stress response, the use of current high-throughput proteomics methods is extremely essential [21,22,23,24]. Acetylation is one of the most common post-translational modifications of proteins [25,26]. Post-translational modification is heavily implicated in plant biological processes and can alter the stability and localization of proteins, affecting their function. For this reason, the analysis of acetylation is extremely important for the understanding of the laws of biological processes [26]. Nevertheless, the critical roles of PSII proteins in acetylation in UV-B reactions are practically unknown.

*Rhododendron chrysanthum* is a species of plant in the family Ericaceae that can be found in the Changbai Mountains in China at an altitude of 1300 to 2650 m; it is a valuable species resource in the world [27]. The annual average temperature at the summit of the Changbai Mountains can drop to 7.3 °C. Due to the harsh climate of the Changbai Mountains, including ultraviolet radiation and low temperatures, *Rhododendron chrysanthum* is resistant to abiotic stresses.

Our previous study reported physiological characteristics of *Rhododendron chrysanthum* after UV-B radiation [27]. Proteomic analysis resulted in the identification of 1395 proteins. The activities of superoxide dismutase (SOD), catalase (CAT), glutathione peroxidase (GPX), and ascorbate peroxidase (APXS) were all significantly elevated in *Rhododendron chrysanthum*, as were the expressions of APXS and GPX. The network of protein interactions showed that antioxidant enzymes play an important role in plant stress resistance [28]. Changes in the amino acid and carbohydrate metabolism of *Rhododendron chrysanthum* under UV-B stress were also reported. Metabolome analysis identified 404 metabolites, in which amino acid content was significantly greater than that of domesticated *Rhododendron chrysanthum* and carbohydrate content was significantly lower than domesticated *Rhododendron chrysanthum*. During the UV-B radiation in *Rhododendron chrysanthum*, the level of transcription of genes related to sucrose and starch metabolism was positively correlated with the content of metabolites, while the metabolism of amino acids was the opposite. The results of this study will provide some implications for the elucidation of the molecular mechanism of UV-B tolerance in plants [29]. However, because our understanding of the effect of UV-B on alpine plants is limited, these results highlight the need for additional research into probable defensive mechanisms of UV-B-irradiated alpine plants.

Throughout this study, we employed *Rhododendron chrysanthum* as materials to investigate its photosynthetic capacity under UV-B radiation. In addition, the effects of UV-B stress on photosystem II proteins of *Rhododendron chrysanthum* were studied by acetylation proteomics and proteomics.

## 2. Materials and Methods

### 2.1. Plant Material and Treatment

*Rhododendron chrysanthum* plants were collected from the Changbai Mountains. They were maintained in an artificial climate room simulating an alpine environment [20]. The plants were grown in an artificial climate room at 18 °C (14-h light)/16 °C (10-h dark) under a 50-μmol (photon) m^−2^ s^−1^ white fluorescent light, with a relative humidity of 60%.

UV-B (280–315 nm) and PAR (400–700 nm) were used in the current study. PAR refers to the light between 400 and 700 nm required to support photosynthesis of plants. Tissue seedlings from two sets of wild-type *Rhododendron chrysanthum* plants were treated with PAR and UV-B for 8 h per day, after 2 days of UV-B stress application—the PAR group serving as the control group (CG) and the UV-B serving as the treatment group (BG). After treatment, 8-month-old BG and CG plants’ leaves were harvested. Proteins that were extracted from the removed leaves were immediately used. Three biological replicates (i.e., six plants) were taken for each group to ensure adequate coverage.

### 2.2. Measurement of Chlorophyll Fluorescence

To measure the induction characteristics of chlorophyll fluorescence in BG and CG leaves, the Imaging-PAM Maxi (Walz, Effeltrich version, Rohrdorf, Germany) was used. The samples were placed in the dark for 20 min before measurements were made. The parameters Fo, Fm, Fv/Fm, NPQ, qP, and ETR were obtained from the data. Analysis of these parameters then allowed us to compare the exact photosynthetic performance.

### 2.3. Protein Extraction

Plants were ground in liquid nitrogen and then transferred to 5-mL centrifuge tubes and sonicated three times on ice with the aid of a high-intensity ultrasound processor (Scientz, Ningbo, China) in a lysis buffer (8 M urea, 2 mM EDTA, 10 mM DTT, and 1% protease inhibitor cocktail). After centrifugation at 20,000× *g* at 4 °C for 10 min, the remainder of the debris was removed. The protein in the supernatant was then centrifuged at 4 °C for 3 min, followed by precipitation with ice-cold 15% TCA at −20 °C for 4 h, and the remaining precipitate was washed with ice-cold acetone three times. Finally, the protein was redissolved in buffer (8 M urea, 100 mM TEAB, and pH 8.0), and the protein concentration in the supernatant was estimated using a 2-D Quant assay kit according to the manufacturer’s instructions.

### 2.4. Trypsin Digestion

The protein solution was reduced with 5-mM dithiothreitol for 30 min at 56 °C and then alkylated with 11-mM iodoacetamide for 15 min at room temperature in the dark for the purpose of digestion. Next, 100-mM TEAB was then added to the diluted protein sample at a urea below 2 M. Finally, trypsin was added at a 1:50 trypsin-to-protein mass ratio for the first digestion overnight and a 1:100 trypsin-to-protein mass ratio for a second 4-h digestion.

### 2.5. LC-MS/MS Analysis and Database Search

The tryptic peptides were dissolved in 0.1% formic acid (solvent A), loaded directly onto a homemade analytical column with an inverted workpiece (15-cm length, 75-μm i.d.). The gradient consisted of an increase from 6% solvent B to 23% solvent B (0.1% formic acid in 98% acetonitrile) over 26 min, a decrease from 23% to 35% over 8 min, and an increase to 80% over the course of 3 min, followed by an 80% associative hold for the final 3 min, all at a continuous flow rate of 400 nL/min on a UPLC 1000 EASY-nLC instrument. A supply of NSI was applied to the peptides followed by tandem mass spectrometry (MS/MS) in Q Exactive TM Plus (Thermo) coupled in line with UPLC. An applied electrospray voltage of 2.0 kV was used in the presence of the conjugate. The *m*/*z* scan range for the total scan was between 350 and 1800, and it is quite possible that intact peptides could be detected inside Orbitrap at 70,000 resolutions. Peptides were then elected for MS/MS analysis victimization in the NCE setting as 28 and, as a result, fragments were detected inside the Orbitrap at a resolution of 17,500, which was a data-dependent procedure that alternated between a MS scan followed by 20 MS/MS scans with a dynamic exclusion of 15.0 s. Auto-gain control (AGC) was set to 5E4. The primary attached mass was set to a value of 100 *m*/*z*.

The resulting MS/MS information was processed using the Maxquant victimization search engine (v.1.5.2.8). Tandem mass spectra were searched against the concatenated human UniProt information concatenated with the reverse decoy information. Trypsin/P was used as a cleavage protein enabling up to four incomprehensible cleavage events to occur. The mass tolerance for precursor ions was set as 20 ppm in the initial search and 5 ppm in the main search, and the fragment particle mass tolerance was set to 0.02 Da. The modifications of carbamidomethyl on Cys were fixed modifications, and modifications of acetylation and oxidization on Met were variable modifications. FDR was adjusted to <1%, and the minimum score for changed peptides was set >40.

For protein identification, the database used was the transcriptome library obtained from the previous research, which contained a total of 45,945 proteins. Modified elements may contain at least one segment of a modified peptide at a site. The FDR for protein identification and PSMs was set to 1%.

### 2.6. Proteomics and Bioinformatics Analysis

Peptides were made by digesting proteins following extraction. The Maxquant search engine (v.1.5.2.8) was used to find the MS/MS information. The UniProt-GOA database was used for the supply for the Gene Ontology comment protein. The protein pathway was annotated victimization of the KEGG database from the Kyoto Encyclopedia of Genes and Genomes. We use WoLF PSORT to predict subcellular location.

### 2.7. Protein Functional Enrichment

#### 2.7.1. GO Enrichment Analysis

There are three main categories of GO annotations of proteins: biological process, cellular component, and molecular function. Significance analysis of the GO enrichment of differentially edited proteins (using the identified protein as the background) was performed using Fisher’s exact test, and a *p* value < 0.05 was considered significant.

#### 2.7.2. KEGG Pathway Enrichment Analysis

KEGG pathway enrichment analysis was performed using the Kyoto Encyclopedia of Genes and Genomes database (KEGG). Significance analysis of enrichment of the KEGG pathway of differentially edited proteins (using the identified protein as the background) was performed using Fisher’s exact test, and a *p* value < 0.05 was considered significant.

#### 2.7.3. Protein Domain Enrichment Analysis

Protein domain enrichment analysis was performed on the InterPro database. Significance analysis of enrichment of protein domains of differentially modified proteins with the identified protein as background was performed using Fisher’s exact test, and a *p* value < 0.05 was considered significant.

### 2.8. Statistical Analysis

First, the samples to be compared were chosen to be pairwise, and the fold change (FC) was calculated as the ratio of the average intensity for each site of modification in two groups of samples. The formula is listed as following: *R* denotes the relative quantitative value of the modification site, *i* denotes the sample, and *k* denotes the modification site.
*FC_A/B,k_ = Mean(R_ik_, i**∈ A)/Mean(R_ik_, i**∈ B)*

In order to calculate the significance of the difference between the groups, a T-test was performed on the relative quantitative value of each modification site in the two sample groups, and the corresponding *p*-value was calculated as the index of significance. A *p*-value < 0.05 was considered significant. The quantitative relative change in value was log2 transformed to make the data conform to a normal distribution. This formula is enumerated as follows:*P_k_ = T.test(Log*2*(R_ik_, i ∈ A), Log*2*(R_ik_, i ∈ B))*

### 2.9. Acetylated Proteins Homology Modeling

For protein identification, we used NCBI BLAST to search for the homologous sequences. Three-dimensional structural models of the acetylated proteins were then created using the comparative protein modeling server SWISS-MODEL.

## 3. Results

### 3.1. Rhododendron chrysanthum Photosynthesis Decreased in the Presence of UV-B Stress

The photosynthesis of *Rhododendron chrysanthum* is significantly impacted by UV-B radiation. The IMAGING-PAM chlorophyll fluorescence imaging system (Heinz Walz, Germany) was used to measure the fluorescence parameters of chlorophyll. Figure 1A shows the Fo, Fm, and Fv/Fm images of *Rhododendron chrysanthum* in two groups. When subjected to UV-B stress, Fm fell in *Rhododendron chrysanthum* leaves (Figure 1B). A significant reduction in the Fv/Fo ratio was observed in response to UV-B stress (Figure 1E). We considered qP and NPQ as additional indicators of the light energy’s utility rate. Under UV-B stress, the reduction in qP and NPQ is a reflection of the decrease in light energy consumption (Figure 1F,G). The parameters Fv and Fm represent the maximum quantum yield for PSII photochemistry. The Fv/Fm ratio decreased during therapy (Figure 1D). In contrast, the ETR under UV-B stress was not significantly altered (Figure 1H). 

### 3.2. Acetylated Proteome in Rhododendron chrysanthum Leaves Responds to UV-B Stress

Proteomic methods were used to compare the abundance of BG and CG proteins from *Rhododendron chrysanthum*. At the time of testing, those with a *p*-value < 0.05 and a differential expression of greater than 1.5-fold were considered to be differentially expressed proteins, resulting in a total of 807 differentially expressed proteins (Figure 2). The abundance level of 450 proteins increased after UV-B stress, and and that of 357 proteins decreased after UV-B stress. In this study, there were 685 acetylated UV-B responsive proteins in *Rhododendron chrysanthum*. From those, 95 proteins demonstrated increased acetylation levels and 590 had reduced acetylation levels.

### 3.3. COG Classification of the Differential Acetylated Proteins in Rhododendron chrysanthum Leaves under UV-B Stress Conditions

The classification of differentially acetylated proteins and differentially expressed proteins was carried out by COG (Cluster of Orthologous Groups of proteins) (Figure 3). The differentially acetylated proteins and the differentially expressed proteins fell into 22 functional categories. Differential acetylated proteins are shown on the right side of Figure 2; *Rhododendron chrysanthum* has 98 differentially acetylated proteins enriched into post-translational modification, protein revenue, and chaperone, accounting for 18.8% of 73 differentially acetylated proteins that are enriched in the transport and metabolism of carbohydrates, representing 14.0% of 66 differentially acetylated proteins enriched in the production and conversion of energy, representing 12.7%. The differentially expressed proteins are expressed on the left-hand side of Figure 2. Here, 85 differentially expressed proteins are enriched in post-translational modification, protein turnover, and chaperones, accounting for 15.9%; 50 differentially expressed proteins are enriched in translation, ribosomal structure, and biogenesis, accounting for 9.3%; and 46 differentially expressed proteins are enriched in carbohydrate transport and metabolism, which accounts for 8.6% of the total proteins.

### 3.4. GO Functional Annotation Analysis of Differential Acetylated Proteins in Rhododendron chrysanthum Leaves under UV-B Stress

GO classification was performed on the differentially expressed and acetylated *Rhododendron chrysanthum* proteins (Figure 4A). GO terms significantly enriched for differentially expressed proteins and acetylated proteins were divided into three broad functional classifications based on the different functions of the GO terms (Figure 4B–D, Table 1). The biological process category consisted of 12 functional subclasses. The number of differentially expressed proteins in the cellular process and the metabolic process was relatively high, 298 (19.9%) and 260 (17.4%), respectively. Differential acetylated proteins also accounted for a high number of functional subclasses in the cell process and the metabolic process, which were as high as 295 (23%) and 280 (21.8%), respectively. There were eight functional subclasses in the category of cell components and a significant number of proteins with differential expression in the four functional subclasses of cells, organelles, membranes, and protein-containing complexes. They were 461 (35.1%), 362 (27.6%), 203 (15.5%), and 97 (7.4%), respectively. The number of differential acetylated proteins was relatively high in the four functional subclasses of cells, membranes, organelles, and polymer complexes, which were as high as 427 (32.6%), 208 (15.9%), 362 (27.7%), and 88 (6.7%). There were six functional subclasses in the molecular function, and differentially expressed proteins accounted for 199 (42.8%) and 188 (40.4%), respectively, in catalytic activity and binding. Differential acetylated proteins accounted for a relatively high number of catalytic activity and binding, which was as high as 255 (51.6%) and 162 (32.8%).

### 3.5. KEGG Enrichment Analysis of Differential Proteins in Rhododendron chrysanthum Leaves under UV-B Stress

For the results of KEGG annotation, 807 DEPs in the sample were mapped to 33 KEGG pathways, and it was found that DEPs were significantly rich in photosynthesis, photosynthesis-antenna protein, and plant pathogen interaction (Figure 5A).

Through the analysis of KEGG enrichment results, the proteins with differential expression were mostly involved in photosynthesis. After detection, the photosynthetic pathway contained 42 distinct proteins, and the proteins of the photosynthetic pathway were analyzed by cluster analysis (Figure 5B). The results showed that the protein of the BG group decreased compared with that of the CG group, indicating that UV-B inhibited the photosynthesis of *Rhododendron chrysanthum*.

Subcellular localization of differentially expressed proteins in the *Rhododendron chrysanthum* photosynthesis pathway (Figure 5C) include chloroplast, cytoplasm, plasma membrane, and mitochondria. We found that 62.5% of the differential proteins were concentrated in the chloroplast and 6.3% in the cytoplasm, among which the number of differential proteins located in the chloroplast was the highest.

According to acetylated proteomics analysis, we found that six differential proteins in photosynthesis were acetylated, and cluster analysis (Figure 5D) was carried out for these six acetylated proteins. The results showed that only “Gene.21753_CL1979.Contig1_All” had a downward trend, while the rest increased. Through localization analysis, we found that three of the six proteins were located in photosystem II, and the rest were located in PSI.

### 3.6. Three-Dimensional Structure Modeling of UV-B Stress-Responsive Acetylated Proteins

In order to gain further insight into the biochemical functions of protein acetylation during the UV-B stress response, the molecular structure of the acetylated proteins was predicted in response to UV-B stress. Three statistically acceptable homology models were built using SWISS-MODEL, with their acetylation sites located within the 3D structure models (Figure 6). PSB27-1, a photosystem II protein, experienced an increase in its acetylation levels under UV-B stress, and the acetylation levels of protein CP47 and CP43 decreased under stress from UV-B. The UV-B stress-increased acetylation site of the PSB27-1 protein occurred on Lys89, the UV-B stress-decreased acetylation site of the CP43 protein occurred on Lys339, and that of the CP47 protein occurred on Lys438 and Lys296. Under UV-B stress, the chloroplast of *Rhododendron chrysanthum* was subjected to acetylation modification of a photosystem II protein (Figure 7), in which the acetylation level of CP43 and CP47 decreased and the acetylation level of PSB27-1 increased.

## 4. Discussion

*Rhododendron chrysanthum* is a valuable plant resource worldwide. Currently, few studies have been conducted on this plant. Most studies, however, are focused on physiology, biochemistry, proteomics, and metabonomics, with few reports on the acetylated proteome. In this experiment, the adaptability of *Rhododendron chrysanthum* to UV-B irradiation conditions was investigated. UV-B tolerance was significant in *Rhododendron chrysanthum*. In this study, *Rhododendron chrysanthum* photosynthesis was found to be inhibited under UV-B stress conditions. Based on acetylated proteomics analysis, the PSII protein was found to be acetylated. PSII protein is thought to be able to render plants resistant to UV-B radiation to some extent and reduce the damage to PSII through modification by acetylation.

In green plants, UV-B radiation can decrease plant photosynthesis [30]. Chlorophyll fluorescence is a sensitive and important indicator of plant photosynthetic traits [31]. In this study, the photosynthetic parameters of *Rhododendron chrysanthum*’s (Fm, Fo, and Fv/Fo) were remarkably reduced as a result of UV-B stress (Figure 1), indicating that the photosystem of the plant was impaired. In contrast, ETR did not change significantly. This suggested that *Rhododendron chrysanthum* may have a few methods to reduce damage to PSII (Figure 8).

For the photosynthetic system, we mainly focused on the research of PSII. Because of its important characteristics of catalytic light-induced cracking and oxygen release, it is often the primary site of damage in environmental stress [31]. Therefore, the structure and function of PSII have been extensively investigated, demonstrating great significance to elucidate the mechanism of light and interactions [32]. In our study, we found that the expression of PSB27, CP47, and CP43 proteins in photosystem II changed under UV-B stress. The acetylation level of PSB27 was up-regulated, while that of CP47 and CP43 were down-regulated.

CP43 and CP47 belong to the nuclear core antenna, which can receive excitation energy from LHCII and CP29, CP26 and other peripheral complexes and feed it to RC. It has been reported that in the process of energy transfer, the peripheral antenna complex first transfers the energy to the CP43, then from CP43 to CP47, and finally from CP47 to the reaction center RC [33]. PSB27 is essential for plants to be able to adapt to changes in light intensity [34]. PSB27 protein is one of the important assembly repair factors that helps PSII maintain good repair and assembly even when it is exposed to stressors such as low temperature, bright light, or a rapid change in light intensity. We believe that PSII stability was improved by UV-B stimulation of substantial protein acetylation. This implies that plant thylakoid membranes experience distinct adaptation processes as a result of UV-B stress. Additionally, the variations in protein acetylation under various environmental circumstances point to a multifunctional role for this alteration in the control and adaptive responses of the photosynthetic machinery. In this study, CP47, CP43, and PSB27 were acetylated. According to the results of cluster analysis, the expression of protein increased, indicating that PSII protein has a certain ability to resist UV-B and reduce the damage of PSII through acetylation modification.

In our study, we identified 945 Lys modification sites on 685 proteins that are involved in a wide range of processes and pathways and can be found in a variety of cellular compartments. The proteins related to photosynthesis are a characteristic group. We found that some differentially expressed proteins in PSII were acetylated, and the acetylation levels were different (Figure 8). These results showed that when plants were under UV-B stress, photosynthesis was inhibited and PSII was damaged, but PSII protein can reduce the damage of PSII to a certain extent through acetylation modification, giving plants a certain ability to resist UV-B.

## 5. Conclusions

PSII acetylated proteins in *Rhododendron chrysanthum* were rarely examined, despite the fact that UV-B-responsive mechanisms in other *Rhododendron chrysanthum* have been extensively studied using proteomics methods [9]. Using proteomics and acetylated proteomics, this study examined UV-B-responsive proteins in chloroplasts in depth, revealing an essential UV-B-responsive mechanism in *Rhododendron chrysanthum* chloroplasts. Based on our research, it was found that both *Rhododendron chrysanthum*’s Fv/Fm and Fv/Fo were drastically reduced by UV-B radiation. Although our results indicate that *Rhododendron chrysanthum* photosynthesis is inhibited, the results of quantitative proteomics and acetylated proteomics indicate that UV-B stress initiates the pathway of photosynthesis, and PSII proteins may be modified by acetylation to alleviate the damage caused by stress, which provides valuable information for studying the molecular mechanism of UV-B resistance of *Rhododendron chrysanthemum*.

## Figures and Tables

**Figure 1 cells-12-00478-f001:**
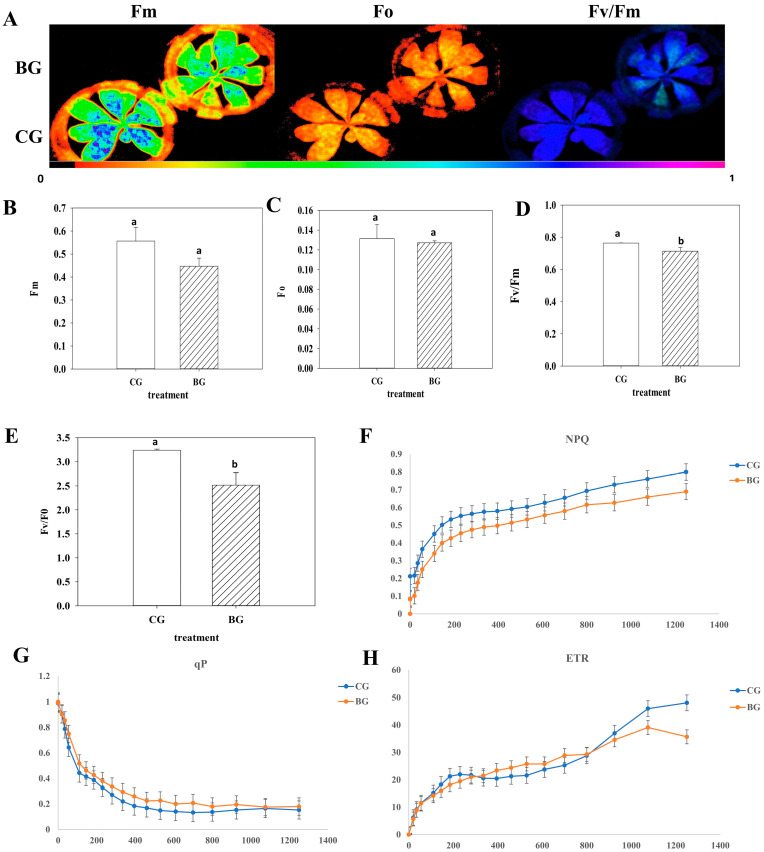
Photosynthetic characteristics in *Rhododendron chrysanthum* under UV-B stress. Lowercase letters denote significant differences between treatments (*p* < 0.05). (**A**) Real-time fluorescence imaging of *Rhododendron chrysanthum*; (**B**) maximal fluorescence (Fm); (**C**) initial minimum fluorescence (**D**) maximum quantum yield of PSII (Fv/Fm); (**E**) potential activity of optical system II (Fv/Fo); (**F**) mean nonphotochemical quenching (NPQ) as a function of PAR; (**G**) mean photochemical quenching (qP) as a function of PAR; (**H**) electron transportation rate (ETR).

**Figure 2 cells-12-00478-f002:**
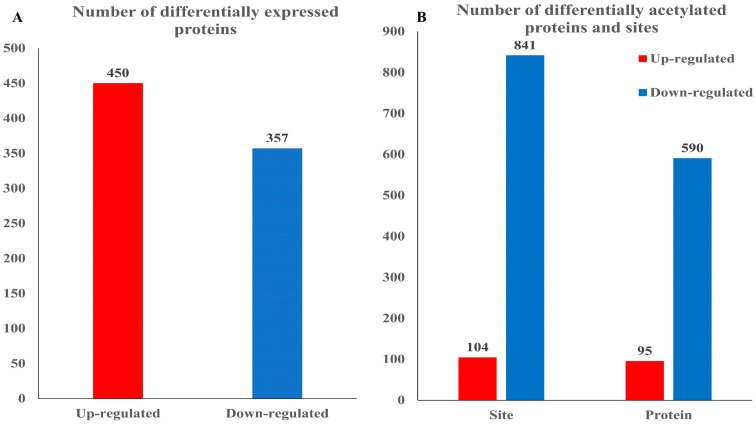
Quantity of differential proteins and acetylated proteins of *Rhododendron chrysanthum* under UV-B stress. (**A**) Number of differentially expressed proteins; (**B**) number of differentially acetylated proteins and sites.

**Figure 3 cells-12-00478-f003:**
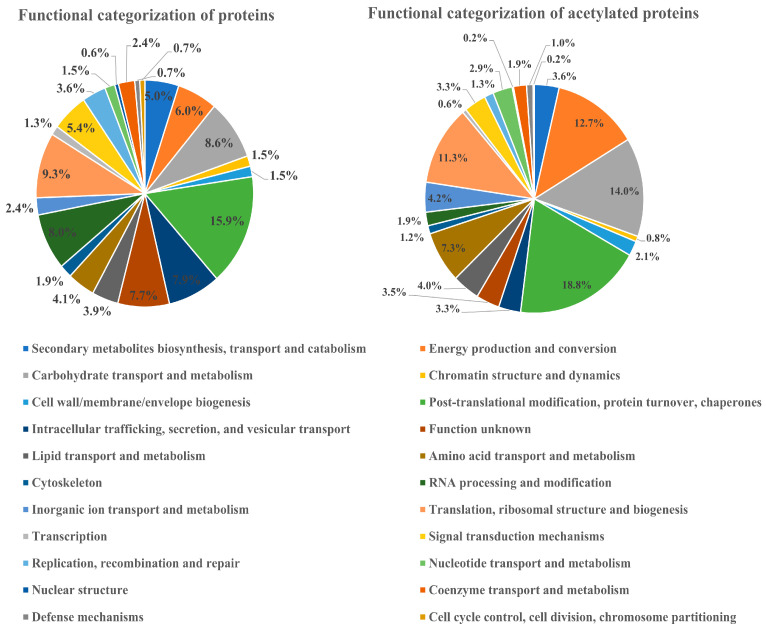
COG classification of differential proteins and differential acetylated proteins under UV-B stress.

**Figure 4 cells-12-00478-f004:**
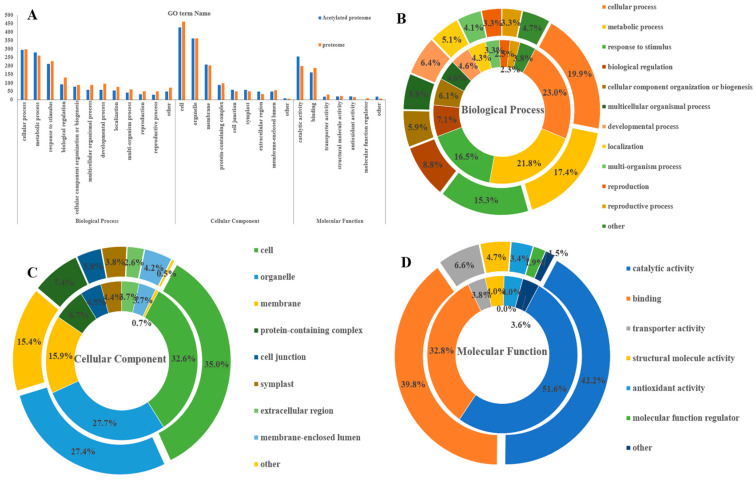
Analysis of differentially acetylated protein and differentially expressed protein GO under UV-B stress. (**A**) The GO terms of differentially protein and differentially acetylated protein; (**B**) The biological process of differentially protein and acetylated protein; (**C**) The cellular component of differentially protein and acetylated protein; (**D**) The molecular function of differentially protein and acetylated protein. The inner circle is differentially acetylated protein and the outer circle is differentially protein.

**Figure 5 cells-12-00478-f005:**
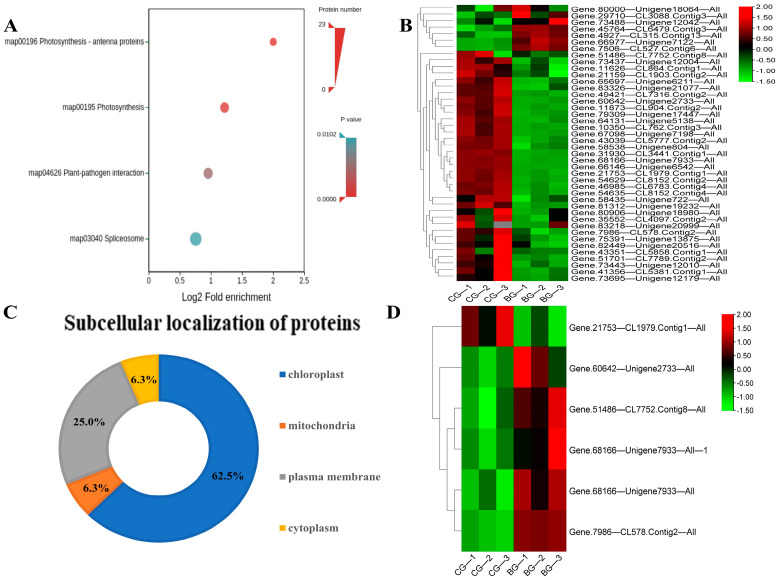
KEGG enrichment analysis, clustering analysis, and subcellular location prediction of the UV-B-responsive proteins. (**A**) KEGG enrichment analysis of differential proteins under UV-B stress; (**B**) heatmap of the photosynthetic proteins; (**C**) subcellular localization of the photosynthetic proteins; (**D**) heatmap of the photosynthetic acetylated proteins.

**Figure 6 cells-12-00478-f006:**
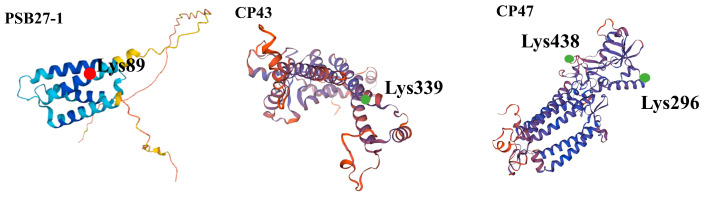
Three-dimensional homologous model of photosystem II acetylated proteins in response to UV-B stress. Red indicates an increase in acetylation level, while green indicates a decrease in acetylation level.

**Figure 7 cells-12-00478-f007:**
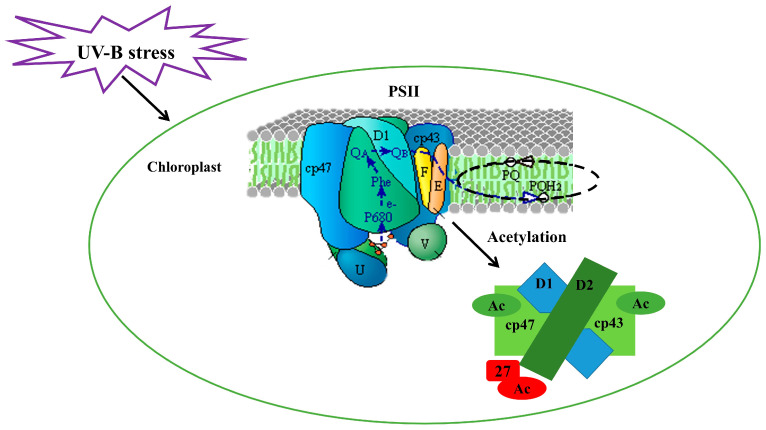
Pattern diagram of photosystem II protein acetylation in *Rhododendron chrysanthum* under UV-B stress.

**Figure 8 cells-12-00478-f008:**
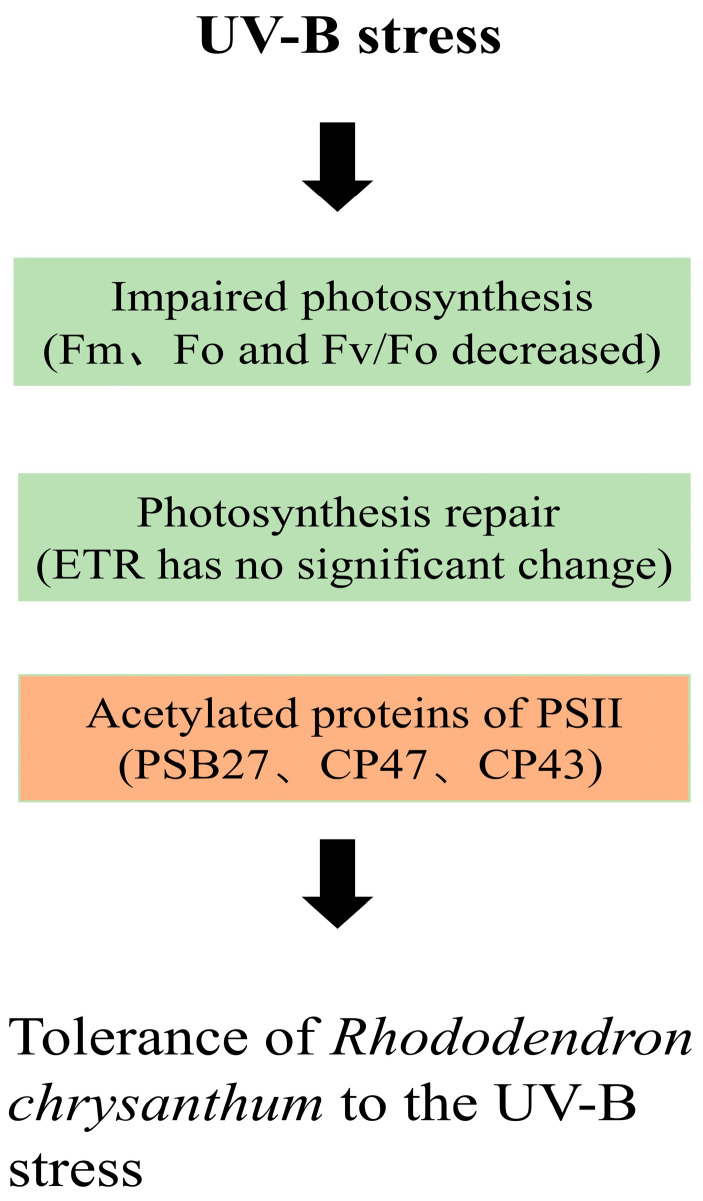
Schematic overview of acetylated proteomic strategy against UV-B stress in *Rhododendron chrysanthum*. Photosynthetic adaptation (green box) and PSII acetylated protein (yellow box) are shown.

**Table 1 cells-12-00478-t001:** GO classification number of differential acetylated protein and differential expressed protein under UV-B stress.

GO Terms Level 1	GO Terms Level 2	Number of Proteins	Number of Acetylated Proteins
Biological Process	Cellular process	298	295
	Metabolic process	260	280
	Response to stimulus	228	212
	Biological regulation	131	91
	Cellular component organization or biogenesis	88	78
	Multicellular organismal process	87	59
	Developmental process	95	59
	Localization	76	55
	Multi-organism process	61	42
	Reproduction	50	32
	Reproductive process	50	30
	Other	71	49
Cellular Component	Cell	461	427
	Organelle	362	362
	Membrane	203	208
	Protein-containing complex	97	88
	Cell junction	50	59
	Symplast	50	58
	Extracellular region	34	49
	Membrane-enclosed lumen	56	48
	Other	0	9
Molecular Function	Catalytic activity	199	255
	Binding	188	162
	Structural molecule activity	22	20
	Antioxidant activity	16	20
	Transporter activity	31	19
	Other	0	18

## Data Availability

The data used in this study are available from the corresponding author on submission of a reasonable request.

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
