# Peer review of "Acetylated Proteomics of UV-B Stress-Responsive in Photosystem II of Rhododendron chrysanthum"

_cells, 2023, doi:10.3390/cells12030478_

Round 1
Reviewer 1 Report
Dear Authors,
The obtained from the authors results are interesting, but in my opinion, the reviewed manuscript needs many improvements and clarifications. However, there are some aspects that are not sufficient to justify and discuss the results obtained.
1. The research question is one of the most important parts of your research. What is the main question addressed by the research?
2. The review of the literature is not thorough so the reader is not given an adequate background about the topic. Authors can improve the introduction part and discuss it.
3. My major concerns are about the methodology. There are deficiencies in the description of the methodology.
4. Explaining some tables in figures can easily express the results (e.g., Figures 4, 6).
5. When you are using abbreviations in tables and figures, all abbreviations used in tables and figures should be defined in the table note or figure caption, respectively, even though the abbreviations will also be defined in the text if they are used there.
6. There are also repetitions in the discussions and in any case, they must be rewritten.
7. Are the conclusions consistent with the evidence and arguments presented? Do they address the main question posed? The arguments and data presented are not complete in establishing the core idea presented.
8. The English in the present manuscript is not of publication quality and requires improvement. Please carefully proofread spell check to eliminate grammatical errors.
In my review, I identified some major weaknesses of the manuscript, mostly linked to methodological aspects in the experimental design and presentation of the results. Therefore, I regret to inform you that your manuscript cannot be accepted for publication as it is. Anyway, as the manuscript is valuable and brings interesting results, I would encourage the authors to carefully revise the manuscript taking into account my suggestions and remarks, in order to definitively improve the quality of the work.
Reviewer 2 Report
The current review article entitled “Acetylated proteomics of UV-B Stress-Responsive in photosystem II of Rhododendron chrysanthum” by Liu et al. is a very interesting study. The manuscript has a good baseline; however, it needs some corrections and adjustments before considering it to be published in Cells.
Abstract: Abstract is written well, quite straightforward. Overall, the English of abstract could be improved to make it clearer.
Introduction
1. Authors should rearrange the data in introduction. Sentence structure of the introduction could be improved with recent references. Sentences are very long and confusing.
2. Second paragraph completely lacks the focus and authors should rewrite it.
3. Authors should try to combine the paragraphs with same type of information in the introduction portion.
4. Give the objective of conducting this study in the last paragraph of introduction.
Materials and Methods:
1. What is PAR?
2. Why the authors selected PAR and UV-B as treatments.
3. Treatment strategy is bit confusing. Please write it more clearly,
4. How many biological and technical replicates were used for this study?
5. Authors need to explain more about the protein identification strategy after LC-MS/MS analysis.
6. Authors must clearly explain the methods of iTRAQ labeling
7. Which databases were used for the protein identification? What was the selection criteria of proteins? How many matched peptides were considered, p-value etc. Authors should explain this in more detail
8. Its surprising to see authors didn’t write anything about the statistical analysis. Haven’t performed or they just missed it?
Results:
9. Figure quality needs to be improved. Figures are very blur, couldn’t see.
10. The authors must justify that why only PSII PTMs have been investigated rather that PSI?
11. Authors must re-make clear figures of gene ontology and KEGG pathway
12. Authors must explain in detail that how acetylation is the only PTM involved in UV-B radiation responsiveness despite of many other different kinds of PTMs.
Discussion:
1. Overall results and discussion section need extensive revisions as there are so many confusions that are misleading.
2. In the discussion section, proper linkage of different references is missing. It seems authors enlist multiple references without proper linkage.
3. There are many problems in the article format and writing. The manuscript needs extensive revisions in terms of English grammar and formatting.
General comments:
4. Use scientific names in italics.
5. Abbreviate the words at the beginning and then use the abbreviations throughout the manuscript.
6. Use homogenous terms for the explanation, don’t use multiple terms for the same purpose.
7. Avoid formatting mistakes.
Reviewer 3 Report
The first sentence of the introduction is no longer correct; reference [1] is not up-to-date. The treatment description "were treated with PAR and UV-B for 48 hours" is utterly inadequate, because (1) such a long irradiation with UV is very unnatural, and (2) fluence rate and spectrum of radiation must be given.
The acronym PTM is introduced multiple times; actually no need to use it, better spell out the few times it is used.
A couple of examples of inadequate language:
line 23 «the photosystem of Rhododendron chrysanthum dropped»
lines 38–39 «Since plants inevitably become exposed to UV-B radiation as a result of inherent fixation»
Round 2
Reviewer 1 Report
Dear Authors,
Thanks for submitting a revised version of your manuscript. Still, there are some aspects that are not sufficient to justify and discuss the results obtained.
. The review of the literature is not thorough so the reader is not given an adequate background about the topic. Authors can improve the introduction part and discuss it.
Reviewer 2 Report
I am surprised to see that the authors replaced the term Post-translational modification with post-translational mutation.
Similarly, stress is replaced with anxiety. Authors need to work on the abstract and modify it according to the previously provided suggestions. Just changing the valid terms with strange ones doesn't fulfill the purpose.
The figures are still not clear. Especially Figures 3, 4, and 5 are very blurry, and it is impossible to see the writing in them.
Reviewer 3 Report
Please see attachment
